# Comparison of Physical Characteristics, Strength and Power Performance Between Elite 3 × 3 and 5 × 5 Male Basketball Players

**DOI:** 10.3390/sports13040090

**Published:** 2025-03-21

**Authors:** Mladen Mikić, Milan Isakov, Nikola Andrić, Alen Ninkov, Aleksandar Karać, Tatjana Jezdimirović Stojanović, Marko D. M. Stojanović

**Affiliations:** 1Faculty of Sport and Physical Education, University of Novi Sad, 21000 Novi Sad, Serbia; mladen.mikic@fsfvns.edu.rs (M.M.); nikola.trenaznaekspertiza18@gmail.com (N.A.); a.karac89@gmail.com (A.K.); 2Training Expertise Laboratory, 21000 Novi Sad, Serbia; ninkov7@gmail.com (A.N.); tatjanaj.ns@gmail.com (T.J.S.); 3University Centre for Intradisciplinary and Multidisciplinary Studies and Research (UCIMSI), University of Novi Sad, 21000 Novi Sad, Serbia; spilsomir@gmail.com

**Keywords:** reactive strength index, countermovement jump, drop jump

## Abstract

**Background/Objectives:** The aim of this study was to explore the differences in physical characteristics, leg strength, and jumping performance between 3 × 3 and 5 × 5 male basketball players. **Methods:** Twelve elite-level 5 × 5 basketball players (26.0 ± 13.0 years; 201.4 ± 6.6 cm; 95.50 ± 11.50 kg) and twelve elite-level 3 × 3 basketball players (26.7 ± 7.3 years; 193.0 ± 5.1 cm; 98.03 ± 9.77 kg), all male, were enrolled in the study. After anthropometric measurements and standardized warm ups, countermovement jump (CMJ), drop jump (DJ) and isokinetic strength testing were conducted, respectively. **Results:** An independent two-sample t-test revealed that 5 × 5 athletes were notably (*p* < 0.005) taller, with a lower body fat percentage (11.9 ± 3.6% vs. 18.6 ± 10.9%) and higher quadricep strength (317.21 ± 36.54 N·m vs. 284.76 ± 29.77 N·m and 313.32 ± 24.08 N·m vs. 285.87 ± 31.2 N·m for left and right leg, respectively). Conversely, 3 × 3 players displayed superior CMJ performance in concentric and eccentric peak forces, peak power, and reactive strength index. In the DJ, 3 × 3 players also excelled in eccentric peak force, reactive strength index, and jump height. **Conclusions:** The findings indicate that while 5 × 5 basketball players excel in body physique and in the strength of their lower body, 3 × 3 basketball players outperform them in power-related metrics.

## 1. Introduction

A popular variation of the traditional 5 × 5 basketball format is a 3 × 3 half-court game. Initially employed primarily as a training method for 5v5 basketball, it has subsequently evolved into a distinct basketball discipline. In 2007, FIBA heightened its emphasis on 3 × 3 basketball, presenting it as an alternative format of the sport. A major milestone was reached a decade later with the announcement of 3 × 3 basketball’s inclusion in the Tokyo 2020 Olympic Games [1]. Today, 3 × 3 basketball stands as an independent sport, with competitions adhering to internationally recognized rules that differ from those of 5v5 basketball. Games of 3 × 3 basketball take place on smaller courts (15 × 11 m compared to 15 × 28 m), involve fewer players (3 vs. 5), and have shorter total playing times (10 min vs. 40–48 min), along with reduced shot clock durations per possession (12 s vs. 24 s) when compared to 5 × 5 basketball games, resulting in unique demands on the court [2,3]. Consequently, there is a growing body of research investigating diverse aspects of 3 × 3 basketball, aiming to optimally develop team training strategies, technical skills, and physical characteristics of players in accordance with the specific demands of the sport. The majority of research has focused on quantifying the demands of 3 × 3 basketball games, evaluating the technical–tactical requirements [3,4], internal load [5,6], and external load [7,8] for junior and adult 3 × 3 players, male and female, competing at national and international levels.

Despite certain assumptions about the similarity of the physical characteristics between 3 × 3 basketball and traditional 5 × 5 play, there is currently insufficient evidence to either support or refute the generalization of findings from one game format to the other [1,6]. Research has indicated [9] that 3 × 3 basketball, played on half of a court, requires greater movement, faster running speeds, and more intense acceleration and deceleration when compared to 5 × 5 basketball. Additionally, Willberg et al. [10] found that players participating in 3 × 3 basketball demonstrate a higher frequency of medium- and high-intensity accelerations, decelerations, jumps, and change-of-direction movements per minute compared to those in 5 × 5 basketball. In contrast, Figueira et al. [11] reported no significant differences in the physiological responses of players between the 3 × 3 and 5 × 5 formats among amateur basketball players. Taken together, these findings suggest a need for further research to identify the factors that contribute to success in 3 × 3 basketball.

Information regarding the physical performance traits of 3 × 3 basketball players is scarce, especially at the elite level of 3 × 3 competition [1,12]. Moreover, the physical characteristics of 3 × 3 basketball players have traditionally been evaluated within the framework of 5 × 5 basketball. Historically, a significant number of 3 × 3 players were individuals who had not achieved prominence in 5 × 5 competition or whose careers in 5 × 5 basketball were already declining [4]. The rapid proliferation of 3 × 3 basketball’s popularity over the last 10–15 years has prompted a growing number of players to specialize in this discipline, thereby facilitating the adoption of structured, systematic, and sport-specific training methodologies for both individuals and teams [13]. These training approaches are likely to produce specific adaptations that may differ significantly from those seen in traditional 5 × 5 basketball. By comparing physical performance traits across these formats, we can better identify the specific attributes that define success in 3 × 3 basketball. This knowledge is vital for coaching staff and talent scouts who seek to cultivate players specifically suited for the demands of 3 × 3 play, moving beyond 5 × 5 standards that may not adequately reflect the requirements for success in the shorter format. For instance, players may require more emphasis on agility and/or aerobic/anaerobic endurance due to the fast-paced nature of 3 × 3 games. In addition, a focused comparison could yield insights on how physical performance traits affect injury prevalence and recovery strategies between formats. Therefore, performing comparative research on 3 × 3 and 5 × 5 basketball is more than just an academic endeavor; it is an important step toward improving player performance and health, refining coaching strategies, and fostering the overall growth of the sport. Interestingly, studies with this aim are scarce, with only one article published recently [14]. The authors compared physical and performance (countermovement vertical jump (CMJ) and sprint) characteristics between 3 × 3 and 5 × 5 top-tier professional male basketball athletes. No significant differences were observed between groups in body mass, height, and age (*p* > 0.05), nor in any concentric or eccentric force-time metrics of countermovement jump. Furthermore, 5 × 5 basketball players attained greater average (−4.01 ± 0.44 m/s^2^ vs. −3.20 ± 0.42 m/s^2^) and maximal decelerations (−6.90 ± 0.93 m/s^2^ vs. −5.61 ± 0.80 m/s^2^) as well as time to stop (1.46 ± 0.18 vs. 1.68 ± 0.18) during sprints. Given the small sample size, the findings of this study cannot be generalized, even among top-level basketball players. Additionally, solely countermovement jump time-force metrics were assessed, whilst it has been shown that other jump modalities such as drop jump can provide additional information about distinct neuromuscular qualities [15] found to be determinant of succes in various team sports, including basketball [16].

Therefore, to address a gap in the scientific literature, this study aimed to investigate the differences in physical characteristics, lower body strength, and concentric/eccentric force-time metrics from countermovement jump and drop jump between elite 3 × 3 and 5 × 5 basketball players.

## 2. Materials and Methods

### 2.1. Study Design/Experimental Approach to the Problem

This study employed a between-subject cross-sectional design with all measurements conducted in a single visit. All measurements were collected during spring 2023/2024 at “Training expertise” laboratory, Novi Sad, Serbia, in the morning hours. The participants were grouped in sets of 6 to ensure an optimal testing flow. The testing for each group of participants lasted about 90 min. Testing was conducted at a minimum of 48 h after a match or high-intensity training. We could not regulate potential differences in chronic fatigue levels—given that 3 × 3 basketball players were in their preparation period while 5 × 5 players were nearing the conclusion of their season, we made efforts to control for acute fatigue by scheduling testing for both groups after a deload week. This approach is substantiated by a recent article by Cabarkapa et al. [17], reporting both concentric and eccentric force-time metrics consistent over the course of a regular season but with some eccentric metrics showing a slight improvement following a short tapering period.

Testing procedures were taken in the following order: (a) anthropometrics and body composition, (b) power tests, and (c) isokinetic strength testing. The tests were supervised by a full professor (M.S.) with extensive experience in strength training and testing (more than 20 years), along with two PhD students. Each tester was responsible for distinct testing areas: the M.S. oversaw anthropometrics and body composition, one PhD student conducted the isokinetic strength testing, and the other PhD student was in charge of the power tests. This structure ensured high quality in the testing procedures.

A standardized warm up, which consisted of lower body activation with mini bands, dynamic stretching exercises with jogging, and landing skills, 10–12 min in duration, preceded power tests. In addition, two sets of three submaximal concentric repetitions of the tested muscle groups preceded the isokinetic testing.

#### Subjects

Professional elite-level 5 × 5 basketball (n = 12; age: 26.0 ± 13.0 years; body height: 201.4 ± 6.6 cm; and body mass: 95.50 ± 11.50 kg; body fat: 11.9 ± 3.6%) and 3 × 3 basketball (n = 12; age: 26.7 ± 7.3 years; body height: 193.0 ± 5.1 cm; and body mass: 98.03 ± 9.77 kg; body fat: 18.6 ± 10.9%) players, all male, were enrolled in this study. A post hoc power analysis (G*power v3.1.9.6, Heinrich-Heine-Universität Düsseldorf, Düsseldorf, Germany) for an alpha level of 0.05 with an effect size of 0.85 (large effect size) and with a sample size of 24 subjects showed a power of 0.51. Both samples in our study included athletes occupying all common positional roles typically found in 3 × 3 and 5 × 5 basketball teams. Therefore, while recognizing that different positions may generally require varying attributes of strength and power, our study reflects the performance metrics across a diverse group of players in each format. Participants in the 3 × 3 basketball competition included members of the Republic of China national team, the Austria national team (currently the European champions in 3 × 3 basketball), and a player from the Serbia national team, who has been repeatedly recognized as the world’s best 3 × 3 player. The basketball participants comprised members of the Vojvodina Basketball Club, which competes in the regional ABA 2 League (where they were the vice-champions) and the Basketball League of Serbia (where they secured the title of regular season champions). Furthermore, the 5 × 5 basketball team consists of one former NBA player and four players who have competed for their respective national teams in the past. Subjects had been performing all testing procedures as part of their regular training and assessments. Inclusion criteria for participation were as follows: (I) no injury in past 4 months; (II) no illness in past 7 days. All participants gave their written and informed consent prior to their enrollment in this study. They were made aware at the time of consent that their participation was voluntary and that they had the option to withdraw from the study at any time. Ethical approval for the study was obtained from the University of Novi Sad Ethical Advisory Committee (Ref. No. 33-01-07/2021-3, approval date 12 March 2023) and adhered to the guidelines set forth in the Declaration of Helsinki.

### 2.2. Testing Procedures

#### Anthropometrics and Body Composition

All anthropometric measurements were performed by an accredited level 2 anthropometrist (M.S.) following the International Society Advancement Kinanthropometry guidelines. The athlete’s height was measured to the nearest centimeter using a portable stadiometer with a sliding head piece (SECA 213, Hamburg, Germany), while their weight was assessed to the nearest tenth of a kilogram with a SECA model scale (Seca GmbH, Hamburg, Germany). Skinfold thicknesses were measured to the nearest 0.1 mm with a Harpenden calipers (West Sussex, UK). The skinfold sites measured include triceps, abdominal, front thigh, and suprailiac [18]. Three measurements were taken at each skinfold site to calculate an average, which was then used for further analysis.

COUNTERMOVEMENT JUMP (CMJ)—The countermovement jump is a reliable and validated test frequently used to assess leg power [19]. Test was conducted using portable force platforms (K-Deltas, Kinvent Inc., Montpellier, France), adhering to the Bosco protocol. Participants began in a standing position, and then moved into a self-selected half squat during the braking phase. Without pausing, they executed a full and explosive vertical extension of the legs through concentric contraction, concluding the test with a landing in an upright position. Participants were instructed to maintain their hands on their hips throughout the test and to minimize the time interval between eccentric and concentric contractions, thereby facilitating an effective coupling of these phases. Each participant completed three trials, with a passive rest period of 45 s interspersed between jumps. The force-time metrics associated with the countermovement jump (CMJ) were selected based on previously published research findings [14,20]. The variables of interest included jump height, concentric peak and mean force, concentric peak and mean power, eccentric peak and mean force, eccentric peak and mean power, and the modified reactive strength index. The optimal performance parameters were recorded for subsequent analysis.

DROP JUMP-DJ—The drop jump is reliable and comonly used test to establish an individual’s reactive strength capacity [21]. The drop jump test was conducted using a portable force plate system (K-Deltas, Kinvent Inc., Montpellier, France). The procedure commenced with the participant standing on a box 40 cm above the force platforms, with their hands positioned on their hips throughout the test. The participant then dropped from the box onto the force platforms, ensuring that both feet landed simultaneously. Upon landing, the participant was instructed to immediately jump as high as possible, aiming to minimize the duration between the eccentric and concentric contractions. Participants completed 3 trials, allowing for 45 s of passive recovery between repetitions. The DJ force-time metrics were selected based on previously published research reports [22,23]. The variables of interest were; concentric peak and mean force, eccentric peak and mean force, reactive strength index and jump height. The best jump performance parameters were selected for further analysis.

ISOKINETIC DYNAMOMETRY—Isokinetic dynamometry is considered the gold standard for evaluating muscle performance in both healthy individuals and clinical populations, backed by numerous studies confirming its reliability and validity [24]. Concentric peak torque for knee extension and knee flexion was measured at angular velocity of zano, 60°/s using the Kineo Training System (Kineo) (V7, GLOBUS, Bolzano, Italy). The Kineo device was calibrated according to the manufacturer’s specifications. Participants were strongly encouraged to exert maximal force and velocity during both knee extension and flexion tasks. Stabilization straps were secured across the trunk, waist, and distal femur of the leg being tested. Testing commenced with knee extension followed by knee flexion, with a 60 s rest period provided between limbs and a 240 s rest period between the flexion and extension assessments. Each participant executed three maximal repetitions for both knee extension and flexion. The length of the lever, defined as the distance from the center of the knee joint proximally to the lateral malleoli of the ankle joint distally, was utilized in the calculation of peak torque. The peak torque was computed using the formula: Peak Force × Lever Length × 9.81 m/s^2^.

KNEE EXTENSION—The test begins with the participant seated, with both the knee and hip flexed at a 90-degree angle. Upon the command of the strength and conditioning (S&C) coach, the participant exerts maximal effort to extend the knee fully, then gradually returns to the starting position. Three trials were conducted for each leg, and the best performance for each velocity was chosen for subsequent analysis.

KNEE FLEXION—The test starts with the participant standing, with the knee and hip fully extended. In response to the S&C coach’s command, the participant pulls their heel towards their hip with maximal force until reaching a 90-degree angle, then slowly returns to the starting position. Three trials were performed for each leg, and the best trial for each velocity was selected for further analysis.

### 2.3. Statistical Analysis

An independent sample *t*-test was conducted to determine the differences in anthropometrics and performance variables between basketball players (n = 12) and 3 × 3 basket players (n = 12). For each *t*-test, the assumption of normality was assessed using the Shapiro–Wilk test, and Levene’s test was used to assess the homogeneity of variances. Cohen’s d effect sizes for were calculated to determine the magnitude of group differences, with magnitudes of effect sizes were assessed using the following criteria: small (0.2), medium (0.5), large (0.8), and very large (1.3) [25]. Statistical significance was set at *p* < 0.05 for all analyses.

## 3. Results

The Shapiro–Wilk test and Levene’s test results for all variables corroborated that the assumptions of normality and equal variance was not violated. Significant differences in body height and body fat percentage were observed between basketball players and 3 × 3 players, with very large effect sizes (Table 1). There were no differences in body weight and age between the two groups.

Significant differences between groups were observed for Quad PT 60° L and Quad PT 60° R, with large effect sizes. In addition, there were no significant differences for Ham PT 60° L and Ham PT 60° R (Table 2).

In terms of concentric countermovement jump (CMJ) performance metrics, signifi-cant differences were observed between groups for both countermovement jump Concentric Peak Force (CMJCPF) and countermovement jump Concentric Mean Force (CMJCMF), with large effect sizes noted (see Table 2). No significant differences were detected for countermovement jump Jump Height (CMJ JH), countermovement jump Concentric Peak Power (CMJCPP), or countermovement jump Concentric Mean Power (CMJCMP).

When examining eccentric CMJ performance metrics, significant differences emerged between groups for Eccentric Peak Force (CMJEPF) and Eccentric Mean Force (CMJEMF), both exhibiting very large effect sizes, as well as for Eccentric Peak Power (CMJEPP) and Reactive Strength Index modified (CMJRSImod), which presented large effect sizes (see Table 2).

No significant differences were found for concentric Drop Jump (DJ) metrics, specifi-cally Drop Jump Peak Force (DJCPF) and Drop Jump Concentric Mean Force (DJCMF). In contrast, among eccentric DJ metrics, DJ Eccentric Peak Force (DJEPF) did reveal a signifi-cant difference between groups with a large effect size, whereas no significant difference was observed for DJ Eccentric Mean Force (DJEMF). Finally, both Drop Jump Reactive Strength Index (DJRSI) and Drop Jump Jump Height (DJ JH) were significantly different between groups, also displaying large effect sizes (see Table 2).

## 4. Discussion

This study investigated differences in physical characteristics, lower body strength, and concentric/eccentric force-time metrics from countermovement jump and drop jump between elite 3 × 3 and 5 × 5 basketball players. The results indicated that 5 × 5 basketball players are noticeably taller and have a significantly lower body fat percentage. Moreover, they exhibit a significantly higher level of quadricep strength, while no difference was observed for hamstring strength. Regarding CMJ force-time metrics, 3 × 3 basketball players exhibit notably higher values in concentric peak and mean force, eccentric peak and mean force, eccentric peak power and modified reactive strength index. No significant differences between groups were observed for jump height, concentric peak and mean power. Regarding DJ force-time metrics, 3 × 3 basketball players displayed significantly higher eccentric peak force, reactive strength index and jump height, while no differences were observed for concentric peak and mean force. Altogether, the results reveal that 5 × 5 basketball players dominate in body physique and lower body strength, while 3 × 3 basketball players are superior in power-related parameters.

When examining anthropometric characteristics, 5 × 5 basketball players exhibited statistically greater body height and a significantly lower body fat percentage, with large effect sizes of 1.877 and 1.527, respectively. When examining anthropometric characteristics, 5 × 5 basketball players exhibited statistically greater body height and a significantly lower body fat percentage, with large effect sizes of 1.877 and 1.527, respectively. It should be acknowledged that significant differences in body fat percentage between players in different formats can influence performance outcomes. A lower body fat percentage can contribute to a higher power-to-weight ratio, crucial fitness attribute in sports like basketball where body weight must be propelled or moved rapidly [26]. In addition, it has been reported that lower body fat percentages were associated with improved performance in jumping tasks [27]. Hovewer, a somewhat contradictory finding in our study was that players with higher levels of body fat achieved better results in power tests, suggesting that the demands of the game format may favor specific fitness profiles, potentially overriding the influence of individual body composition. We acknowledge that future research could explore how body composition interacts with different game formats to provide deeper insights into performance dynamics. Understanding the relationship between body composition and game format could enhance our strategies for athlete preparation and performance optimization.

Our findings are not in line with a recently published study [14], which reported no statistical difference and small effect size for body height between 3 × 3 and 5 × 5 elite basketball players. In addition, while 3 × 3 players body height obtained in our study is in line with recently published data in similar cohort [14], body height of 5 × 5 players reported in our study is higher than previously reported. For example, the average heights of NCAA Division-I basketball players were found to be 193 ± 0.8 cm [28] and 196.4 ± 11.9 cm [29], while an average height of 199.5 ± 8.2 cm was reported for elite Serbian basketball players [30]. Altogether, it seems that that the specific sample of 5-on-5 basketball players in our study, with among the highest body height values published, is responsible for the observed difference in body height between the two groups. Considering body fat in 5 × 5 basketball players, the obtained results are in line with a recently presented reference values for male basketball players body fat percentage measured via skinfold (12.4%; 95% CI 10.6–14.2%) [31]. As far as we know, this is the first study to assess the body fat percentage of elite 3 × 3 basketball players. The results, showing an average of 18.6% ± 5.93, indicate that a more advantageous body composition profile (with lower fat mass) could be beneficial, given that higher body fat levels have been linked to decreased power-related performance in basketball, including changes of direction [32] and vertical jumps [33].

Lower body strength is an important predictor of basketball performance [34]. For example, lower body strength has been shown to be a strong predictor of change of direction ability [35], sprint and basketball specific jumps [36] as well as playing time [37]. Finally, it has been showed that elite basketball players achieved significantly better performances in knee extensors peak torques (+20.2%, *p* ≤ 0.05) as compared to average-level players [38]. The results of our study showed that 5 × 5 basketball players display both quads (317.21 ± 36.54 N·m) and hamstring (284.76 ± 29.77 N·m) strength levels that fall in the range of previously reported data for quads (297.5 ± 43.5 N·m) and hamstring strength (172.0 ± 27.3 N·m) in elite male basketball players [39]. In addition, similar values (289 ± 3 N·m and 157 ± 8 N·m for quads and hamstring strength, respectively) were recently presented for a cohort of elite male basketball players [40], corroborating our study findings. There is a paucity of research concerning lower body strength performance in 3 × 3 basketball players, particularly among those competing at professional levels. To the best of our knowledge, this is the first study reporting isokinetic leg strength in 3 × 3 elite players, so no comparison with previous investigations can be made. In addition, Cabarkapa et al. [12] recently reported 1RM back squat of elite 3 × 3 players to fall within the previously reported ranges for NCAA Division-I collegiate players, implying similar lower body strength levels between two cohorts. However, the results of the current study indicate that 5 × 5 basketball athletes exhibited significantly greater quadricep strength (p=0.026 and *p* = 0.025 for the left and right legs, respectively), with large effect sizes. In addition, there were no significant differences in hamstring strength, with p-values of 0.852 and 0.837 for the left and right legs, respectively. Collectively, our study findings demonstrate that elite 3 × 3 basketball players display significantly lower quadriceps strength while showing comparable hamstring strength in relation to 5 × 5 basketball players. Absolute values of 3 × 3 basketball players align with previously reported results at the elite basketball level. Therefore, we suggest that both groups possess adequate levels of lower body strength. However, further research with larger sample sizes is needed to confirm or dispute our findings.

Since both 3 × 3 and 5 × 5 basketball players perform over 50 jumps and frequently accelerate, decelerate, or change direction every few seconds during active gameplay, with the intensity and frequency increasing over time [2,30], it is clear that the significance of eccentric muscle actions and stretch-shortening cycle ability becomes increasingly vital for achieving success on the court. It has been evidenced that players who possess superior eccentric and stretch-shortening muscle qualities exhibit more efficient gameplay and ultimately secure more playing time [41]. Authors found a significant positive relationship (r = 0.406 to r = 0.552) between eccentric mean force, eccentric mean and peak power, and playing efficiency in professional male basketball players.

Our study identified significant differences favoring 3 × 3 basketball players across various sport-specific jumping performance measures, with effect sizes varying from large to very large (0.85 to 1.57). Notably, the most significant differences (very large effect sizes) were observed in the eccentric phases of Countermovement Jump (CMJ) and Drop Jump (DJ) movements, specifically in eccentric peak and mean force. Additionally, large effect sizes were noted for the CMJ and DJ Reactive Strength Indexes and DJ Jump Height, suggesting that 3 × 3 basketball players distinguished to 5 × 5 players primarily by their superior eccentric strength, eccentric power and stretch-shortening cycle capabilities. Our study findings confront recently presented results by Cabarkapa et al. [14] on comparable sample and with similar study design. They reported no statistically significant differences between 3 × 3 and 5 × 5 basketball players in any eccentric or concentric force-time metrics of the countermovement jump. We can argue that top level sub-sample of 5 × 5 basketball players, consisting of NBA and EuroLeague players compared to our basketball subsample (ABA 2 League) is likely responsible for differences between two study findings. Indeed, it has been shown that as the competitive level increases, fitness attributes also improve, particularly those associated with dynamic strength parameters [42]. Consequently, it seems reasonable to assume that superior fitness profile of 5 × 5 basketball players in Cabarkapa et al. [14] study is responsible for obtaining comparable CMJ force-time metrics results between 3 × 3 and 5 × 5 basketball players. More research about the topic seems prudent.

It can be hypothesized that observed differences obtained in our study results from difference in 3 × 3 and 5 × 5 game demands and consequent training program specificity. It has been reported that high-intensity requirements during live time are greater in 3 × 3 than 5 × 5 basketball [43,44]. In addition, elite 3 × 3 players perform more change of direction, accelerations and decelerations per minute of game compared to 5 × 5 basketball players, with more frequent high intensities (>3.5 m·s^2^) for all three movement patterns [2]. Considering game duration, 3 × 3 games feature an intermittent structure with frequent high-intensity eccentric, concentric and stretch-shortening cycle actions per minute of game, underlining the need for well-structured training highly oriented to develop these specific neuromuscular abilities. Although reasonable, this statement is highly speculative and most likely could serve as a starting point for future research.

Our study offers insights that could inform practices and policies aimed at enhancing athlete health. By elucidating the distinct physical demands and performance outcomes of 3 × 3 and 5 × 5 basketball players, coaches and trainers can develop specialized training regimens tailored to each format. For example, the superior power-related performance metrics observed in 3 × 3 players suggest that training programs should prioritize these attributes to adequately prepare athletes for the unique requirements of their game. Furthermore, the enhanced eccentric force capacities of 3 × 3 players suggest the necessity for specialized training modalities aimed at improving eccentric strength, as well as focused injury prevention strategies. Such strategies could better equip players to handle the significant eccentric forces encountered during gameplay, thereby promoting overall athlete health. Finally, delineating the distinct characteristics of 3 × 3 and 5 × 5 basketball players can increase public awareness regarding the specific adaptations and impacts of playing 3 × 3 on health-related fitness parameters.

While providing practitioners additional insight about distinction in physical and performance characteristics between elite 3 × 3 and 5 × 5 professional male basketball players, this study is not without limitations. First, like with many studies involving elite athletes, small sample size is likely the most evident one. Future research should include larger and more homogenous cohort of 3 × 3 players. Second, while. we assessed strength and power characteristics of 3 × 3 and 5 × 5 basketball players, it seems prudent to link these findings with some other practical basketball performance metrics. Previous research indicated positive relationship between these attributes and change of direction ability [45], deceleration ability [46] and sprint speed [47]. Establishing these relationships in our specific cohort would enrich the interpretation of our results, further delineate 3 × 3 and 5 × 5 players’ physical characteristics and provide actionable insights for coaches and athletes aiming to optimize performance. Finally, at the time of testing, groups were in different parts of the season (preseason for 3 × 3 basketball and finish of competitive season for 5 × 5 basketball group), which potentially could influence study findings. Future studies should gather data in comparable part of the seasons.

In conclusion, the results of this study indicate that elite professional male basketball players in 3 × 3 and 5 × 5 formats exhibit different physical characteristics, strength, and power performance. The results revealed that 5 × 5 basketball players are significantly taller and have a leaner physique, along with a higher level of quadriceps strength. In contrast, 3 × 3 players exhibited greater reactive and explosive strength, with eccentric force-time metrics being the main factor contributing to these outcomes. These findings could help sports scientists and practitioners in the athlete selection process and aid in creating specialized training programs. 

## Figures and Tables

**Table 1 sports-13-00090-t001:** Anthropometric characteristics and comparison statistics for 3 × 3 and 5 × 5 basketball players. The values with * indicate statistically significant differences between groups (*p* < 0.05).

Parameter	Basketball Players		3 × 3 Players		*t*	*p*	Cohen’s d
	M	SD	M	SD			
Body height (cm)	201.41	5.03	193.00	3.84	4.598	0.001 *	1.877
Body weight (kg)	95.50	6.05	98.03	6.25	−1.028	0.315	0.420
Body fat (%)	11.90	1.84	18.60	5.93	−3.741	0.001 *	1.527
Age (years)	26.08	6.08	26.75	2.92	−0.342	0.735	0.140

**Table 2 sports-13-00090-t002:** Strength parameters, countermovement and drop jump force-time metrics and comparison statistics for 3 × 3 and 5 × 5 basketball players. The values with * indicate statistically significant differences between groups (*p* < 0.05).

Parameter	Basketball Players		3 × 3 Players		*t*	*p*	Cohen’s d
	M	SD	M	SD			
Quad PT 60° L (N·m)	317.21	36.54	284.76	29.77	2.385	0.026 *	0.974
Quad PT 60° R (N·m)	313.32	24.08	285.87	31.22	2.412	0.025 *	0.985
Ham PT 60° L (N·m)	154.78	28.06	152.78	23.76	0.188	0.852	0.077
Ham PT 60° R (N·m)	153.48	15.27	162.93	33.38	−0.882	0.387	0.360
CMJ JH (cm)	41.22	5.12	41.83	6.47	−0.255	0.801	0.10
CMJCPF (N)	250.97	19.43	284.02	35.39	−2.836	0.010	1.15
CMJCMF (N)	200.96	14.52	219.25	19.89	−2.572	0.017	1.05
CMJCPP (W)	5368.5	712.0	5707.7	782.4	−1.111	0.279	0.45
CMJCMP (W)	3093.6	318.7	3293.4	542.7	−1.099	0.284	0.44
CMJEPF (N)	238.45	20.66	284.63	31.18	−4.277	0.001	1.74
CMJEMF (N)	183.86	12.69	217.66	24.35	−4.301	0.001	1.75
CMJEPP (W)	2059.58	316.44	2577.91	568.85	−2.758	0.011	1.12
CMJRSImod	0.95	0.17	1.13	0.22	−2.139	0.044	0.87
DJCPF (N)	497.29	84.4	493.50	121.8	0.089	0.930	0.03
DJCMF (N)	339.18	39.86	336.78	50.81	0.129	0.898	0.05
DJEPF (N)	542.15	73.36	642.92	144.46	−2.154	0.042	0.88
DJEMF (N)	307.45	35.99	341.98	56.73	−1.781	0.089	0.72
DJ RSI	2.19	0.37	2.63	0.61	−2.088	0.049	0.85
DJ JH (cm)	35.08	4.88	39.58	5.59	−2.097	0.048	0.85

Legend: Quad PT 60° (L/R)—quadriceps Peak Torque 60° (left/right leg); Ham PT 60° (L/R)—hamstring Peak Torque 60° (left/right leg); CMJ JH—countermovement jump height; CMJCPF—countermovement jump concentric peak force; CMJCMF—countermovement jump concentric mean force; CMJCPP—countermovement jump concentric peak power; CMJCMP—countermovement jump concentric mean power; CMJEPF—countermovement jump eccentric peak force; CMJEMF—countermovement jump eccentric mean force; CMJEPP—Countermovement Jump eccentric peak power; CMJRSImod—countermovement jump reactive strength index, modified; DJCPF—drop jump concentric peak force; DJCMF—drop jump concentric mean force; DJEPF—drop jump eccentric peak force; DJEMF—drop jump eccentric mean force; DJ RSI—drop jump reactive strength index; DJ JH—drop jump height; *—*p* < 0.05.

## Data Availability

The data are available upon reasonable request.

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
