# Peer review of "Comparison of Physical Characteristics, Strength and Power Performance Between Elite 3 × 3 and 5 × 5 Male Basketball Players"

_sports, 2025, doi:10.3390/sports13040090_

Round 1
Reviewer 1 Report
Comments and Suggestions for Authors
The article shows interesting research results on the possibility of influencing the selection of the proper participants for a given sport.
So far, it has not been possible to obtain sufficient answers as to what psychomotor and physical characteristics are optimal for 5x5 and 3x3 basketball players.
The world's top 5x5 and 3x3 basketball players were gathered for the study. The study was conducted on the same day with all participants.
The results show that 5x5 basketball players dominate in physique and lower body strength, while 3x3 basketball players are superior in power-related parameters.
The 5x5 basketball players are significantly taller and have a leaner build, as well as higher degrees of quadriceps strength. In contrast, 3x3 players showed greater reactive strength and speed of power generation, with eccentric force time ratings being the main contributor to these results. These findings can help both researchers and sports professionals in the athlete selection process and help in the development of targeted and optimal training programs.
Author Response
please check attach file

Reviewer 2 Report
Comments and Suggestions for Authors
- Line 23. I would suggest "and in the strength of their lower body" since
"lower body strength" could be interpreted as having less body strength. - Line 73. Is this research by the authors published? If so, it should be cited. If not, could it be cited as a working paper?
- Line 104 I would suggest specifying 5x5 for the first group since you specify 3x3 for the second group.
- Line 106 18.6
- Line 110 and nearby. When signing the informed consent papers were the participants told that they might be identifiable from the study sample descriptions? If not, is that an ethical violation?
- Line 197. The items under the Cohen's d column do not seem to line up well.
- Lines 198-200. Does this belong underneath Table 2 rather than Table 1?
- Looking at Table 2, it would appear that effect sizes had to be about 0.85 (large) to be significant statistically. This issue of power might be mentioned as part of the analysis plan.
- Lines 231-232 seem to be repeated (with more detail) at lines 243-244.
- Line 246. I suggest "with a recently"
- Line 273 I suggest "corroborating our study findings"
- Lines 332-343 These implications are very useful, possibly could be stressed with more detail.
- 13. References are not consistent in style.
Reviewer 3 Report
Comments and Suggestions for Authors
The aim of the study to investigate the differences in physical characteristics, lower body strenght, and concentric/eccentric force-time metrics from countermovement jump and drop jump between elite 3×3 and 5×5 basketball players. It is a research that, despite not providing too many variables and not having much novelty, is methodologically well thought out, the manuscript is well written and it is an interesting and enjoyable read.
Here are my contributions:
- Line 11, The aim of the study…
- The paragraph beginning on line 58 should be split in two. At least separate the objective from the study.
- Line 104, in which positions did the basketball players evaluated play? it won't be the same to choose a pibot as a point guard, will it? since it is such a small sample, this can condition the results a lot.
- In lines 133 and 211 the same acronym is referred to but with different meanings and is repeated.
- Lines 331, 344, 363,… unnecessary spaces appear between paragraphs.
Author Response
see attach file

Reviewer 4 Report
Comments and Suggestions for Authors
This review highlights major and minor weaknesses in the manuscript, providing detailed comments with line numbers for necessary improvements.
- Lines 10-15: The study aims to compare 3×3 and 5×5 basketball players, but the introduction does not clearly establish the need for this comparison. While differences in game format are acknowledged, there is no mention of specific gaps in the literature that this study aims to address.
- Lines 50-60: Prior studies on game intensity and movement demands in 3×3 vs. 5×5 basketball are cited, but no references to previous research examining physical and strength differences between these players are included.
- Lines 12-14, 105-110: The sample size is limited to 12 players per group, reducing statistical power and the ability to generalize findings.
- Lines 90-98: Several key factors that may have influenced results are not controlled, including:
- Training background differences (3×3 players may focus more on power training).
- Fatigue levels (5×5 players were tested at the end of the season, 3×3 players in preseason).
- Positional roles (guards vs. centers naturally differ in strength and power).
- The study does not account for how these variables might have impacted group strength and power differences.
- Lines 110-113: The 3×3 sample consists of national-level players, while the 5×5 sample consists of regional-level players, introducing a talent bias that may explain observed differences rather than the game format itself. The difference in the competitive level of the groups significantly influenced all performance measurements, making direct comparisons between groups unreliable.
- Lines 112-120: The study does not mention whether inter-rater reliability was assessed for the strength and power tests, which may introduce measurement variability. There is no reference to previously validated protocols for these tests
- Lines 140-150 (Table 1): The results show significant differences in body height and body fat percentage between 3×3 and 5×5 players, with substantial effect sizes. These anthropometric differences likely significantly influenced the results of the power and strength tests. The differences in player body composition and physiological profiles may explain much of the variance in test outcomes rather than the game format itself. The study does not account for these differences in statistical analyses, leading to potential bias in interpreting the impact of playing format on physical performance.
- Lines 182-190: The study uses independent sample t-tests, which assume normality and equal variance. However, the small sample size raises concerns about whether these assumptions were met.
- Lines 185-187: The Shapiro-Wilk test is mentioned, but the results are not reported, making it unclear whether non-parametric alternatives (e.g., Mann-Whitney U test) should have been used.
- Lines 270-285: While the study assesses strength and power characteristics, it does not examine whether these differences translate into game performance metrics (e.g., agility, sprint speed, shooting efficiency).
- Lines 290-310: The discussion suggests that differences in physical characteristics influence game performance, but without in-game performance data, these claims remain speculative.
Author Response
see attach file

Round 2
Reviewer 4 Report
Comments and Suggestions for Authors
- Based on a thorough review of the authors' responses, the manuscript's changes are insufficient in several critical areas. Here’s my detailed opinion:
- The reviewer pointed out that the introduction inadequately establishes the need for comparing 3×3 and 5×5 basketball formats. While the authors mentioned they addressed this gap (lines 58-79), they did not convincingly clarify why such a comparison is essential beyond merely acknowledging format differences. More explicitly articulating specific gaps in the current literature or practical implications is still required.
- The reviewer rightly criticized the lack of references on differences in training backgrounds between 3×3 and 5×5 athletes. Although the authors cited new references (lines 72-79), they did not adequately integrate this information into the rationale for their study design, leaving the reader questioning the study’s scientific foundation.
- The reviewer identified critical uncontrolled factors: training background, fatigue levels, and positional roles. The authors acknowledged these but argued that their study is descriptive rather than predictive. Their rationale referencing Bishop (2008) is a valid theoretical point. However, admitting significant uncontrolled variables (training specificity, fatigue) without adjusting the statistical analysis significantly weakens the paper’s methodological rigor.
- The authors clarified testing times and the sequencing of measurements, citing a qualified supervisor. They addressed concerns regarding inter-rater reliability and provided validation references for tests used (Countermovement jump, Drop jump, Isokinetic dynamometry). However, they still neglected explicitly discussing inter-rater reliability or reporting measurement validation within their specific.
- The reviewer highlighted that those significant anthropometric differences (height and body fat %) were not accounted for statistically, potentially skewing results interpretation. The authors acknowledged this, proposing the interaction between body composition and game format for future research. However, they failed to adjust their current analysis or adequately discuss how these differences might bias present findings.
- The authors stated they confirmed normality and homogeneity of variance through Shapiro-Wilk and Levene's tests, yet they did not report specific test results, as the reviewer explicitly requested.
- The authors acknowledged the limitation but did not offer any tangible steps to link their strength/power findings with practical basketball performance metrics (sprint speed, shooting efficiency). Their response merely suggests future research rather than enriching the interpretation of the current results.
Author Response
please check attach. file
